

# Evidence of cryptic methane cycling and non-methanogenic methylamine consumption in the sulfate-reducing zone of sediment in the Santa Barbara Basin, California

Sebastian J.E. Krause[1]*[#], Jiarui Liu[1], David J. Yousavich[1], DeMarcus Robinson[2], David W. Hoyt[3], Qianhui Qin[4], Frank Wenzhoefer[5,6,7], Felix Janβen[5,6], David L. Valentine[8], and Tina Treude[1,2]*

[1]Department of Earth Planetary and Space Sciences, University of California, Los Angeles, CA 90095, USA

[2]Department of Atmospheric and Ocean Sciences, University of California, Los Angeles, CA 90095, USA

[3]Pacific Northwest National Laboratory Environmental and Molecular Sciences Division, Richland, WA 99352, USA

[4]Interdepartmental Graduate Program in Marine Science, University of California, Santa Barbara, CA 93106, USA

[5]HGF-MPG Group for Deep-Sea Ecology and Technology, Alfred-Wegener-Institute, Helmholtz-Center for Polar and Marine Research, Am Handelshafen 12, 27570 Bremerhaven, Germany

[6]Max Planck Institute for Marine Microbiology, Celsiusstrasse 1, 28359 Bremen, Germany

[7]Department of Biology, DIAS, Nordcee and HADAL Centres, University of Southern Denmark, 5230 Odense M, Denmark

[8]Department of Earth Science and Marine Science Institute, University of California Santa Barbara, Santa Barbara, CA 93106, USA

*Correspondence: Sebastian Krause (sjkrause@ucsb.edu), Tina Treude (ttreude@g.ucla.edu)



**# Present address: Earth Research Institute, 6832 Ellison Hall, University of California**
**Santa Barbara, Ca 93106-3060**



**Abstract.** The recently discovered cryptic methane cycle in the sulfate-reducing zone of marine
and wetland sediments couples methylotrophic methanogenesis to anaerobic oxidation of
methane (AOM). Here we present evidence of cryptic methane cycling activity within the
upper regions of the sulfate-reducing zone, along a depth transect within the Santa Barbara
Basin, off the coast of California, USA. The top 0-20 cm of sediment from each station was
subjected to geochemical analyses and radiotracer incubations using $^{35}S\text{-}SO_4^{2-}$, $^{14}C$-mono-
methylamine, and $^{14}C\text{-}CH_4$ to find evidence of cryptic methane cycling. Methane
concentrations were consistently low (~3 to ~16 μM) across the depth transect, despite AOM
rates increasing with decreasing water depth (from max 0.05 nmol cm$^{-3}$ d$^{-1}$ at the deepest station
to max 1.8 nmol cm$^{-3}$ d$^{-1}$ at the shallowest station). Porewater sulfate concentrations remained
high (~23mM to ~29 mM), despite the detection of sulfate reduction activity from $^{35}S\text{-}SO_4^{2-}$
incubations with rates up to 134 nmol cm$^{-3}$ d$^{-1}$. Metabolomic analysis showed that substrates
for methanogenesis (i.e., acetate, methanol and methylamines) were mostly below the detection
limit in the porewater, but some samples from the 1-2 cm depth section showed non-
quantifiable evidence of these substrates, indicating their rapid turnover. Estimated
methanogenesis from mono-methylamine ranged from 0.2 nmol to 0.5 nmol cm$^{-3}$ d$^{-1}$.
Discrepancies between the rate constants (K$_1$) of methanogenesis (from $^{14}C$- mono-
methylamine) and AOM (from either $^{14}C$- mono-methylamine-derived $^{14}C\text{-}CH_4$ or from
directly injected $^{14}C\text{-}CH_4$) suggest the activity of a separate, concurrent metabolic process
directly metabolizing mono-methylamine to inorganic carbon. We conclude that the results
presented in this work show strong evidence of cryptic methane cycling occurring within the
top 20 cm of sediment in the Santa Barbara Basin.  The rapid cycling of carbon between
methanogenesis and methanotropy likely prevents major build-up of methane in the sulfate-
reducing zone. Furthermore, our data suggest that methylamine is utilized by both
methanogenic archaea capable of methylotrophic methanogenesis and non-methanogenic





microbial groups. We hypothesize that sulfate reduction is responsible for the additional
methylamine turnover but further investigation is needed to elucidate this metabolic activity.



## *1. Introduction*

In anoxic marine sediment, methane is produced by microbial methanogenesis in the last step of organic carbon remineralization (Stephenson and Stickland, 1933; Thauer, 1998; Reeburgh, 2007). This methane is produced by groups of obligate anaerobic methanogenic archaea across the Euryarchyota, Crenarchaeota, Halobacterota , and Thermoplasmatota phyla (Lyu et al., 2018). Methanogens can produce methane through three different metabolic pathways, using $CO_2$ ($CO_2$ reduction; e.g., hydrogenotrophic) (Eq. 1), acetate (acetoclastic) (Eq. 2) and methylated substrates such as, methyl sulfides, methanol, and methylamines (methylotrophic) (e.g., Eq. 3).

$$4H_2 + CO_2 \rightarrow CH_4 + 2H_2O \qquad\qquad [1]$$

$$CH_3COO^- + H^+ \rightarrow CO_2 + CH_4 \qquad\qquad [2]$$

$$4CH_3NH_2 + 2H_2O \rightarrow 3CH_4 + CO_2 + 4NH_4 \qquad\qquad [3]$$

Classically, hydrogenotrophic and acetoclastic methanogenesis are dominant in deeper sulfate-free sediment (Jørgensen, 2000; Reeburgh, 2007). This distinct geochemical zonation is due to the higher free energy gained by sulfate-reducing bacteria within the sulfate reduction zone coupling sulfate reduction with hydrogen and/or acetate consumption in sulfate-rich sediment. Thus, sulfate-reducing bacteria tend to outcompete methanogenic archaea for hydrogen and acetate in shallower sediment layers in the presence of sulfate (Kristjansson et al., 1982; Winfrey and Ward, 1983; Lovley and Klug, 1986; Jørgensen, 2000). However, methylotrophic methanogenesis is known to occur within the sulfate-reducing zone. The activity of this process in the presence of sulfate reduction is possible because methylated substrates, such as methylamines, are non-competitive carbon sources for methanogens (Oremland and Taylor, 1978; Lovley and Klug, 1986; Maltby et al., 2016; Zhuang et al., 2016; 2018; 2018; Krause and Treude, 2021). Methylotrophic methanogenesis activity in the sulfate-reducing zone has been detected in a wide range of aquatic environments, such as coastal wetlands (Oremland et al., 1982; Oremland and Polcin, 1982; Krause and Treude, 2021),





upwelling regions (Maltby et al., 2016), and eutrophic shelf sediment (Maltby et al., 2018; Xiao
et al., 2018). Despite methylotrophic activity in the sulfate-reducing zone, methane
concentrations are several orders of magnitude lower than methane concentrations found in
deeper sediment zones where sulfate concentrations are depleted (Barnes and Goldberg, 1976;
Dale et al., 2008b; Wehrmann et al., 2011; Beulig et al., 2018).
In anoxic marine sediment, anaerobic oxidation of methane (AOM) is an important
methane sink that is typically coupled to sulfate reduction (Eq. 4) and mediated by a consortium
of anaerobic methane-oxidizing archaea (ANME) and sulfate-reducing bacteria (Knittel and
Boetius, 2009; Orphan et al., 2001; Michaelis et al., 2002; Boetius et al., 2000; Hinrichs and
Boetius, 2002; Reeburgh, 2007).
$CH_4 + SO_4^{2-} \rightarrow HCO_3^- + HS^- + H_2O$ [4]
AOM occurring in the sulfate-reducing zone, fuelled by concurrent methylotrophic
methanogenesis activity, i.e., the cryptic methane cycle, could be the reason why methane
concentrations are consistently low in sulfidic sediment (Krause and Treude, 2021; Xiao et al.,
2017; Xiao et al., 2018). These studies highlight the importance of the cryptic methane cycle
on the global methane budget. However, the extent of our knowledge of cryptic methane cycle
is restricted to a few aquatic environments. Thus, it is crucial to investigate and understand the
cryptic methane cycle in myriad environments to fully understand its impact on the global
methane budget. In the present study we focus on organic-rich sediment below oxygen-
deficient water in the Santa Barbara Basin (SSB), California.
Oxygen minimum zones (OMZ) are regions where high oxygen demand in the water
column leads to a dramatic decline or even absence of dissolved oxygen (Wright et al., 2012;
Paulmier and Ruiz-Pino, 2009; Wyrtki, 1962; Canfield and Kraft, 2022). In these
environments, coastal upwelling of nutrients results in high phytoplankton growth, greatly
enhancing organic matter loading and in turn creating a high metabolic oxygen demand during
organic matter degradation in the water column. This enhanced respiration depletes oxygen



faster than it is replenished (especially in poorly ventilated water bodies), which results in
strong seasonal or continuous low oxygen conditions (Wyrtki, 1962; Helly and Levin, 2004;
Wright et al., 2012; Levin et al., 2009). Sediment beneath OMZs is typically rich in organic
matter supporting predominantly or exclusively anaerobic degradation processes, including
methanogenesis (Levin, 2003; Rullkötter, 2006; Middelburg and Levin, 2009; Fernandes et al.,
2022; Treude, 2011). Thus, sediments underlying OMZ's are good candidate environments to
investigate cryptic methane cycling.

Located within the Pacific Ocean, between the Channel Islands and the mainland of

Santa Barbara, California, USA, the SBB is characterized as a thermally stratified, coastal
marine basin with a maximum water column depth of approximately 590 m (Soutar and Crill,
1977; Arndt et al., 1990; Sholkovitz, 1973). Low oxygen concentrations (<10 µM)  are found
in the bottom waters below the sill depth (~475 m) of the SBB (Sholkovitz, 1973; Reimers et
al., 1996). The sediment in the SBB have an organic carbon content between 2-6%
(Schimmelmann and Kastner, 1993). These characteristics makes the SBB a prime study site
to find evidence of cryptic methane cycling.

Organic carbon sources for methylotrophic methanogenesis, such as methylamine, are

ubiquitous in the coastal marine environments (Zhuang et al., 2018; Zhuang et al., 2016; Oren,
1990), including marine environments where OMZ's exist (Ferdelman et al., 1997; Gibb et al.,
1999). Methylamines are derived from osmolytes, such as glycine and betaine, and are
synthesized by phytoplankton (Oren, 1990). However, the abundance of methylamines and
how they may be driving cryptic methane cycling in anoxic sediment within OMZ's are
virtually unknown. Furthermore, the fate of methane from methylotrophic methanogenesis in
the sulfate reduction zone is poorly constrained. Particularly, if cryptic methane cycling is
active above the sulfate-methane transition zone, gross production and consumption of
methane have likely been underestimated. Therefore, finding evidence for the cryptic methane



cycle in the SBB is a necessary step towards understanding how carbon is cycled through the
sediment of the SBB and other OMZs.
In the present study we report biogeochemical evidence of cryptic methane cycling in
surface sediment (top ~15 cm) collected along a depth transect crossing the SBB. We applied
the radiotracer method from Krause and Treude (2021) to trace the production of methane from
mono-methylamine, followed by the anaerobic oxidation of methane to inorganic carbon. We
combined this approach with standard radiotracer methods for the detection of AOM and
sulfate reduction as well as with analyses of sediment porewater geochemistry.



## *2.  Methods.*
### *2.1. Study site and sediment sampling*
Sediment samples were collected during the R/V *Atlantis* expedition AT42-19 in fall
2019. Collection was achieved with polycarbonate push cores (30.5 cm long, 6.35 cm i.d.),
which were deployed by the ROV *JASON* along a depth transect through the SBB. The depth
transect selected for this particular study, was the Northern Deposition Transect 3 (NDT3),
with three stations (NDT3-A, -C and -D), as well as the Northern Depositional Radial Origin
(NDRO), and the Southern Depositional Radial Origin (SDRO) station, located in the deepest
part of the basin.  Details on the station's water column depths and near-seafloor oxygen
concentrations are provided in Table 1.
**Table 1.** Water column depth, bottom water oxygen concentrations and coordinates of each station sampled during
this study.

| Station | Depth (m) | Bottom Water Oxygen (µM) | Latitude | Longitude |
|---|---|---|---|---|
| SDRO | 586 | 0 | 34.2011 | -120.0446 |
| NDRO | 580 | 0 | 34.2618 | -120.0309 |
| NDT3-A | 572 | 9.2 | 34.2921 | -120.0258 |
| NDT3-C | 498 | 5 | 34.3526 | -120.0160 |
| NDT3-D | 447 | 8 | 34.3625 | -120.0150 |


After sediment collection, ROV push cores were returned to the surface by an elevator
platform. Upon retrieval onboard the R/V *Atlantis,* sediment samples were immediately
transported to an onboard cold room (6°C) for further processing of biogeochemical parameters
(see details in section 2.2.).

### *2.2. Sediment porewater sampling and sulfate analysis*
For porewater analyses, two ROV sediment push cores from each station were sliced
in 1-cm increments in the top 10 cm of the sediment, followed by 2-cm increments below.





During sediment sampling, ultra-pure argon was flushed over the sediment to minimize
oxidation of oxygen sensitive species. The sliced sediment layers were quickly transferred to
argon-flushed 50 mL plastic centrifuge vials and centrifuged at 2300 X g for 20 mins to extract
the porewater. Subsequently, 2 mL of porewater was subsampled from the supernatant and
frozen at -20 °C for shore-based sulfate analysis by ion chromatography (Metrohm 761)
following (Dale et al., 2015). Additional porewater (1 mL) was subsampled for the
determination of the concentration of methylamine and other metabolic substrates (see section

2.4).


### 170   *2.3. Sediment methane and benthic methane flux analyses*

Methane concentration in the sediment was determined from a replicate ROV pushcore.
Sediment was sliced at 1-cm increments in the top 10 cm, followed by 2-cm increments below.
Two mL of sediment was sampled with a cut-off 3 mL plastic syringe and quickly transferred
to 12 mL glass serum vials filled with 5 mL 5% (w/w) NaOH solution. The vials were sealed
immediately with a grey butyl rubber stopper and aluminum crimps, shaken thoroughly, and
stored upside down at 4 °C. Methane concentrations in the headspace were determined shore-
based using a gas chromatograph (Shimadzu GC-2015) equipped with a packed Haysep-D
column and flame ionization detector. The column was filled with helium as a carrier gas,
flowing at 12 mL per minute and heated to 80 °C. Methane concentrations in the environmental
samples were calibrated against methane standards (Scotty Analyzed Gases) with a $\pm$ 5%
precision.
To determine methane flux out of the sediment and into the water column, 1-2
custom-built cylindrical  benthic flux chambers (BFC) (Treude et al., 2009) were deployed at
each sampling station by the ROV Jason. The BFCs consist of a lightweight fiber-reinforced
plastic frame, which holds a cylindrical polycarbonate chamber. Buoyant syntactic foam was
attached to the feet of the frame to keep the BFC's from sinking too deep into the soft and



poorly consolidated sediments, especially in the deeper stations. Water overlying the
enclosed sediment was kept mixed with a stirrer bar rotating below the lid of the chamber.
The BFC's were equipped with a syringe sampler holding 7, 60 mL glass syringes (6 syringes
for sample collection and 1 syringe for freshwater injection). One sample syringe withdrew
50 mL of seawater from the chamber volume at pre-programed time intervals. The seventh
syringe was used to inject 50 mL of de-ionized water into the chamber shortly after
deployment to calculate the volume from the change in salinity in the overlying seawater
recorded by a conductivity sensor (type 5860, Aanderaa Data Instruments, Bergen, NO),
according to (Kononets et al., 2021).

Prior to BFC seawater sample collection, the 26 mL serum bottles were acid washed,

rinsed three times using MilliQ filtered DI water, and then combusted at 300 °C. One to two
pellets of solid NaOH were added into each empty combusted serum bottle. All empty serum
bottles were then flushed with ultra-pure nitrogen gas (Airgas Ultra High Purity Grade
Nitrogen, Manufacturer Part #:UHP300) for 5 min, then sealed with autoclaved chlorobutyl
stoppers and crimps. Lastly, a vacuum pump was used to evacuate the bottles to a pressure
down to <0.05 psi prior to sample collection.

Immediately after BFC recovery from the seafloor, approximately 20 mL of seawater

sample was transferred into the pre-evacuated 26 mL glass serum bottles through the
chlorobutyl stopper using a sterile 23G needle. Pressure within the serum bottle was
equalized to atmospheric pressure with the introduction of UHP grade nitrogen. Serum
bottles were shaken to dilute the NaOH pellets, which terminated metabolic activity and
forced the dissolved methane into the gas headspace. The serum bottles were reweighed after
sample collection, to calculate the exact volume of the seawater sample. Methane
concentrations in seawater collected from the BFC's were analyzed shipboard by gas
chromatography according to Qin et al., 2022.





212   Total methane concentration in the headspace was calculated following the ideal gas

213 law Eq. (5),

214  $n = \frac{PV}{RT} * [CH_4] * \frac{1}{V_{SW}}$ .            [5]

215 Where $n$ is the total molar concentration of methane, $P$ is atmospheric pressure, $V$ is the volume

216 of the headspace of serum bottle (which is calculated by 26 mL subtracted by the volume of

217 seawater sample), $R$ is the ideal gas constant, $T$ is temperature in Kelvin (288.15 K), $[CH_4]$ is

218 the methane measured by GC as percentage values in ppm, and $V_{SW}$ is the volume of seawater

219 in the serum vial. The volume of sampled seawater in each serum bottle was calculated by

220 subtracting the mass of the empty serum bottle from the mass of the filled serum bottle,

221 normalized by the density of seawater.

223 *2.4. Porewater metabolomic analysis*

224   To obtain sediment porewater concentrations of methanogenic substrates

225 (methylamine, methanol, and acetate), 1 mL porewater was extracted from 1-2 cm and 9-10

226 cm depth sections at each station (see section 2.2) and syringe-filtered (0.2 μm) into pre-

227 combusted (350 °C for 3 hrs) amber glass vials (1.8 mL), which were then closed with a PTFE

228 septa-equipped screw caps and frozen at -80 °C until analyses. Samples were analysed at the

229 Pacific Northwest National Laboratory, Environment and Molecular Sciences Division for

230 metabolomic analysis using proton nuclear magnetic resonance (NMR). Prior to analysis,

231 porewater samples were diluted by 10% (v/v) with an internal standard (5 mM 2,2-dimethyl-

232 2-silapentane-5-sulfonate-d6). All NMR spectra were collected using an 800 MHz Bruker

233 Avance Neo (Tava), with a TCl 800/54 H&F/C/N-D-05 Z XT, and an QCl H-P/C/N-D-05 Z

234 ET extended temperature range CryoProbe. The 1D 1H NMR spectra of all samples were

235 processed, assigned, and analysed by using the Chenomx NMR Suite 8.6 software with

236 quantification based on spectral intensities relative to the internal standard. Candidate

237 metabolites present in each of the complex mixture were determined by matching the chemical



shift, J-coupling, and intensity information of experimental NMR signals against the NMR
signals of standard metabolites in the Chenomx library. The 1D 1H spectra were collected
following standard Chenomx data collection guidelines, employing a 1D NOESY presaturation
experiment (noesypr1d) with 65536 complex points and at least 4096 scans at 298 K. Signal to
noise ratios (S/N) were measured using MestReNova 14 with the limit of quantification equal
to a S/N of 10 and the limit of detection equal to a S/N of 3. The 90° $^1$H pulse was calibrated
prior to the measurement of each sample with a spectral width of 12 ppm and 1024 transients.
The NOESY mixing time was 100 ms and the acquisition time was 4 s followed by a relaxation
delay of 1.5 s during which presaturation of the water signal was applied. Time domain free
induction decays (72114 total points) were zero-filled to 131072 total points prior to Fourier
transform.

### 250    *2.5. Metabolic activity determinations*

One replicate ROV sediment push core (hereafter 'ROV rate push core') from each
station was sub-sampled with three mini-cores (20 cm long, 2.6 cm i.d.) for radiotracer
incubations according to the whole-core injection method (Jørgensen 1978) to collect
quantitative metabolic evidence (sulfate reduction, methanogenesis, methane oxidation) of
cryptic methane cycling. The incubation methods are detailed below.

### 257    *2.5.1. Sulfate reduction via $^{35}$S-Sulfate*

Within the same day of collection, one mini-core from each ROV rate push core was
used to determine sulfate-reduction rates. Radioactive carrier-free $^{35}$S-sulfate ($^{35}$S-SO$_4^{2-}$;
dissolved in MilliQ water, injection volume 10 µL, activity 260 KBq, specific activity 1.59
TBq mg$^{-1}$) was injected into the mini core at 1-cm increments and incubated at 6 °C in the dark
following (Jørgensen, 1978). Injected sediment cores were stored vertically and incubated for
~6 hrs at 6 °C in the dark. Incubations were stopped by slicing the sediment in 1-cm increments



into 50 mL plastic centrifuge tubes containing 20 mL 20% (w/w) zinc acetate solution. Each
sediment sample was sealed and shaken thoroughly and stored at -20 °C to halt metabolic
activity. For the control samples, sediments were added to zinc acetate solution prior to
radiotracer injection. In the home laboratory, sulfate reduction rates were determined using the
cold-chromium distillation method (Kallmeyer et al., (2004).

### 2.5.2. Methanogenesis and AOM via $^{14}$C-Mono-Methylamine

This study aimed at determining the activity of methanogenesis from mono-
methylamine (MG-MMA) and the subsequent anaerobic oxidation of the resulting methane to
inorganic carbon by AOM (AOM-MMA). To accomplish this goal, a mini core from each ROV
rate push core was injected with radiolabeled $^{14}$C-mono-methylamine ($^{14}$C-MMA; dissolved in
1 mL water, injection volume 10 µL, activity 220 KBq, specific activity 1.85-2.22 GBq mmol$^{-1}$
$^{1}$) similar to section 2.5.1. After 24 hrs, the incubation was terminated by slicing the sediment
at 1-cm increments into 50 mL wide mouth glass vials filled with 20 mL of 5% NaOH. Five
killed control samples were prepared by transferring approximately 5 ml of extra sediment
from each station into 50 mL wide mouth vials filled with 20 mL of 5% NaOH prior to
radiotracer addition. Sample vials and vials with killed controls were immediately sealed with
butyl rubber stoppers and aluminium crimps and shaken thoroughly for 1 min to ensure
complete biological inactivity. Vials were stored upside down at room temperature until further
processing. In the home laboratory, methane production from $^{14}$C-MMA by MG-MMA and
subsequent oxidation of the produced $^{14}$C-methane ($^{14}$C-CH$_4$) by AOM-MMA was determined
according to the adapted radiotracer method outlined in (Krause and Treude, 2021).
To account for $^{14}$C-MMA potentially bound to mineral surfaces (Wang and Lee, 1993,
1994; Xiao et al., 2022), we determined the $^{14}$C-MMA recovery factor (RF) for the sediment
from the stations NDT3-C, D and NDRO according to Krause and Treude (2021).



Metabolic rates of MG-MMA were calculated according to Eq. 8. Note that natural

concentrations of MMA in the SBB sediment porewater were either below detection or
detectable, but below the quantification limit (<10 µM) (Table S1). Therefore, MMA
concentrations were assumed to be 3 µM to calculate the ex-situ rate of MG-MMA (Eq. 8).
$$MG\text{-}MMA = \frac{a_{CH_4} + a_{TIC}}{a_{CH_4} + a_{TIC} + \left[\frac{a_{MMA}}{RF}\right]} * [MMA] * \frac{1}{t} \qquad [7]$$
where *MG-MMA* is the rate of methanogenesis from mono-methylamine (nmol cm$^{-3}$ d$^{-1}$); $a_{CH4}$
is the radioactive methane produced from methanogenesis (CPM); $a_{TIC}$ is the radioactive total
inorganic carbon produced from the oxidation of methane (CPM); $a_{MMA}$ the residual
radioactive mono-methylamine (CPM); RF is the recovery factor (Krause and Treude, (2021)
; *[MMA]* is the assumed mono-methylamine concentrations in the sediment (nmol cm$^{-3}$); *t* is
the incubation time (d). $^{14}$C-CH$_4$ and $^{14}$C-TIC sample activity was corrected by respective
abiotic activity determined in killed controls.

Results from the $^{14}$C-MMA incubations were also used to estimate the AOM-MMA

rates according to Eq. 8,
$$AOM\text{-}MMA = \frac{a_{TIC}}{a_{CH_4} + a_{TIC}} * [CH_4] * \frac{1}{t} \qquad [8]$$
where *AOM-MMA* is the rate of anaerobic oxidation of methane based on methane produced
from MMA (nmol cm$^{-3}$d$^{-1}$); $a_{TIC}$ is the produced radioactive total inorganic carbon (CPM); $a_{CH4}$
is the residual radioactive methane (CPM); *[CH$_4$]* is the sediment methane concentration (nmol
cm$^{-3}$); *t* is the incubation time (d). $^{14}$C-TIC activity was corrected by abiotic activity determined
by replicate dead controls.

### 310  *2.5.3 Anaerobic oxidation of methane via $^{14}$C-Methane*

AOM rates from $^{14}$C-CH$_4$ (AOM-CH$_4$) were determined by injecting radiolabeled $^{14}$C-

CH$_4$ (dissolved in anoxic MilliQ, injection volume 10 µL, activity 5 KBq, specific activity
1.85–2.22 GBq mmol$^{-1}$) into one mini core from each ROV rate core at 1-cm increments similar



to section 2.5.1. Incubations of the mini cores were stopped after ~24 hours similar to section
2.5.2. In the laboratory, AOM-CH$_4$ was analysed using oven combustion (Treude et al., 2005)
and acidification/shaking (Joye et al., 2004). The radioactivity was determined by liquid
scintillation counting. AOM-CH$_4$ rates were calculated according to Eq. 8.

### 319  *2.5.4 Rate constants for AOM-CH$_4$, MG-MMA, and AOM-MMA*

Metabolic rate constants (k) for AOM-CH$_4$, MG-MMA and AOM-MMA were
calculated using the experimental data determined by sections 2.5.2 and 2.5.3. The rate
constants consider the metabolic reaction products, divided by the sum of reaction reactants
and products and by time. The metabolic rate constants for AOM-CH$_4$, MG-MMA and AOM-
MMA were calculated according to Eq. 9,
$$k = \frac{a_{products}}{a_{products} + a_{reactants}} * \frac{1}{t} \qquad [9]$$
where $k$ is the metabolic rate constant (day$^{-1}$); $a_{products}$ is the radioactivity (CPM) of the
metabolic reaction products; $a_{reactants}$ is the radioactivity (CPM) of the metabolic reaction
reactants; $t$ is time in days.



### *3.   Results*

### *3.1. Sediment biogeochemistry*

At most stations, porewater methane concentrations in the top 10-20 cm of sediment fluctuated between 3 and 13 µM with no clear trend (Fig. 1A, E, I, M, and Q). At NDRO, methane steadily increased below 12 cm, reaching 16 µM at 14–15 cm (Fig. 1E).  Methane concentrations determined in water samples from the BFC incubations revealed only minor fluctuations over time with no clear trends suggesting no net fluxes of methane into or out of the sediment at all stations (Fig.1S). It is notable, however, that the BFCs captured higher methane concentrations (350-800 nM) in the supernatant of station SDRO, NDRO, and NDT3-A compared to NDT3-C and NDT3-D (< 130 nM). Sulfate concentrations showed no strong decline with depth at any station (except maybe a weak tendency at SDRO and NDT3-A) and fluctuated between 23 and 30 mM in the sampled top 10-20 cm (Fig. 1A, E, I, M, and Q).

Table S1 provides porewater concentrations of organic carbon sources from the metabolomic analysis, as measured by NMR, that are known to support methanogenesis. Methylamine was detected at SDRO and NDT3-A (1–2 cm), but those concentrations were below the quantification limit (10 µM). Otherwise, methylamine was below detection (<3 µM) for all other samples. Similarly, methanol was detected but below quantification at NDT3-A (1–2 cm) but otherwise below detection. Acetate was at a quantifiable level (21 µM) at NDT3-A (1–2 cm) but was otherwise either below quantification (SDRO, 1-2 cm; NDRO, 1-2 cm) or below detection.

### *3.2 AOM from $^{14}$C-methane and sulfate reduction from $^{35}$S-sulfate*

Fig. 1B, F, J, N, and R depicts ex-situ rates of AOM-CH$_4$ and sulfate reduction from the radiotracer incubations with $^{14}$C-methane and $^{35}$S-sulfate in sediment mini cores, respectively. AOM-CH$_4$ activity tended to increase with decreasing water depth in the top 5 cm of the sediment (from max 0.05 nmol cm$^{-3}$ d$^{-1}$ at NDRO to max 4.5 nmol cm$^{-3}$ d$^{-1}$ at NDT3-



D), while rates were either negligible (SDRO, NDRO, NDT3-A) or <1 nmol cm$^{-3}$ d$^{-1}$ (NDT3-
C, NDT3-D) for depths >5 cm. Where peaks in AOM were present (SDRO, NDT3-C, NDT3-
D) they were always located in the top 0–1 cm sediment layer.
Sulfate reduction activity was detected throughout all sediment cores with the highest
rates mostly at 0–1 cm, followed by a decrease with increasing sediment depth. The highest
individual sulfate reduction peaks were found at NDRO, NDT3-A, and NDT3-C (120, 85 and
133 nmol cm$^{-3}$ d$^{-1}$). At NDT3-D sulfate reduction rates varied between 14 and 45 nmol cm$^{-3}$ d$^{-}$
$^{1}$ throughout the core with no clear trend. Note that sulfate reduction data are missing for 0–5
cm at SDRO. Here, rates gradually decreased from 52 to 10 nmol cm$^{-3}$ d$^{-1}$ below 5 cm.

**Figure 1.** Depth profiles of biogeochemical parameters in sediment across the depth transect of the Santa Barbara Basin. A, E, I, M, and Q: sediment methane and porewater sulfate; B, F, J, N, and R: AOM-CH$_4$ and sulfate reduction (determined from direct injection of $^{14}$C-CH$_4$ and $^{35}$S-Sulfate, respectively); C, G, K, O, and S: AOM-MMA and MG-MMA (determined from direct injection of $^{14}$C-MMA); D, H, L, P, and T: rate constants for AOM-CH$_4$, MG-MMA and AOM-MMA.



### 3.3 Methanogenesis and AOM from $^{14}C$-mono-methylamine

### 3.3.1 $^{14}C$-MMA recovery from sediment



RF values determined in sediments from NDRO, NDT3-C and D stations (see section
2.5.2) were 0.93, 0.84, and 0.75, respectively. They were used to correct MG-MMA rates at
each station of the study. Note that no RF values were determined for SDRO or the NDT3-A.
We applied RF values from NDRO and NDT3-C, respectively, instead.

### 3.3.2 MG-MMA and AOM-MMA


Fig. 1C, G, K, O, S shows ex-situ rates of MG-MMA and AOM-MMA, assuming a
natural MMA concentration of 3 μM (see section 2.5.2). At SDRO, NDRO, and NDT3-A, MG-
MMA ranged between 0.27 and 0.45 nmol cm$^{-3}$ d$^{-1}$ throughout the sediment core without trend
(Fig. 1C, G, and K).  At NDT3-C MG-MMA ex-situ rates were lower ranging between 0.007
nmol cm$^{-3}$ d$^{-1}$ and 0.3 nmol cm$^{-3}$ d$^{-1}$ without any pattern (Fig. 1O). At NDT3-D, MG-MMA
sharply increased from 0.05 nmol cm$^{-3}$ d$^{-1}$ at 0–1cm, to ~0.34 nmol cm$^{-3}$ d$^{-1}$ at 1–2 cm. MG-
MMA then decreased slightly to ~0.2 nmol cm$^{-3}$ d$^{-1}$ between 2 and 9 cm, before increasing to
~0.5 nmol cm$^{-3}$ d$^{-1}$ at the bottom of the core (Fig. 1S).
AOM-MMA rates were 1 to 2 orders of magnitude higher than MG-MMA rates and 1
to 4 orders of magnitude higher than AOM-CH$_4$ rates (Fig 1C, G, K, O, S). At SDRO, NDRO,
NDT3-A, and NDT3-C, AOM-MMA ex-situ rates ranged between 5.3 and 10 nmol cm$^{-3}$ d$^{-1}$
(unless zero) with no trend (Fig 1C, G, K, and O). At NDT3-D, AOM-MMA rates decreased
from 15.9 nmol cm$^{-3}$ d$^{-1}$ at 1–2 cm to 9 nmol cm$^{-3}$ d$^{-1}$ at 11–12 cm (Fig. 1S). At all stations,
some sediment intervals showed no biological net AOM-MMA activity (Fig 1C, G, K, O, S).
In these sediment intervals, the $^{14}C$-TIC activity was statistically not different from the average
plus the standard deviation of the killed control samples.

### 3.4 Rate constants for MG-MMA, AOM-MMA and AOM-CH$_4$




Fig. 1D, H, L, P, and T show the rate constants (k) for MG-MMA, AOM-MMA and

AOM-CH$_4$ for the comparison of relative radiotracer turnover. At all stations, MG-MMA rate
constants were between 0.01 and 0.15 d$^{-1}$. AOM-CH$_4$ rate constants ranged between 0.0009 d$^{-}$
$^1$ and 0.3 d$^{-1}$. Rate constants for AOM-MMA, however, were considerably higher than MG-
MMA and AOM-CH$_4$ with values ranging between 0.7 and 1.2 d$^{-1}$. Most rate constants
remained constant over depth, with the exemption of AOM-MMA at station NDT3-C and D
(Fig. 1P and T), which showed a steady decrease below 9 cm.



### *4. Discussion*

### *4.1. Evidence of cryptic methane cycling*

The aim of the present study was to check for the existence of cryptic methane cycling in SBB surface sediments by presenting evidence for the concurrent activity of sulfate reduction, AOM, and methanogenesis through radiotracer incubations ($^{35}$S -SO$_4^{-2}$, $^{14}$C-CH$_4$, and $^{14}$C-MMA, respectively). Our study confirmed indeed that the three processes co-exist at all investigated stations (Fig. 1). The most prominent concurrent metabolic activity was evident from activity peaks near the sediment-water interface at station NDT3-C (Fig. 1N and O). We suggest the concurrent peaking was stimulated by the availability of fresh, i.e., recently deposited, organic matter coinciding with low oxygen concentrations in the bottom water (Table 1). Fresh organic material likely provided a source for both organoclastic sulfate reduction and methylotrophic methanogenesis, and indirectly (i.e., linked to the methane produced) for AOM coupled to either nitrate, iron, or sulfate reduction. Low oxygen concentrations offered favourable conditions for anaerobic processes in the surface sediment. At the remaining stations (SDRO, NDRO, SDT3-A, SDT3-D; Fig. 1), metabolic activity of all three processes was also confirmed near the sediment surface (with the exemption of the missing data for sulfate reduction at SDRO), but they not always depicted rate peaks (particularly not for AOM-CH$_4$).

Methane detected in the sulfate-rich sediment (Fig. 1A, E, I, M, Q) was likely produced by methylotrophic methanogenesis utilizing non-competitive substrates within the sulfate-reducing zone (Oremland and Taylor, 1978; King et al., 1983; Maltby et al., 2016; Maltby et al., 2018; Reeburgh, 2007), which is also indicated by the production of methane from our $^{14}$C-MMA incubations. It is interesting to note that methane concentrations remained relatively constant around 5 to 12 µM while AOM-CH$_4$ tended to increase with decreasing water depth. This pattern suggests that the threshold partial pressure of methane (the Michaelis constant K$_m$)



of AOM remained at steady state between AOM and methanogenesis (compare, e.g., with
Conrad 1999).
The finding of relatively constant methane concentrations in surface sediments is
against the general view that methane concentrations above the sulfate-methane transition zone
show a linear, diffusion-controlled decline towards the sediment-water interface, where
methane escapes into the water column (Reeburgh, 2007). We argue that the non-linear
methane trends we observe in the present study is an indication for simultaneous methane
production and consumption, i.e., cryptic methane cycling, as evident from our radiotracer
experiments.
As  methanogenesis activity showed considerable activity even at the sediment-water
interface (0-1 cm) at all stations, aside from station NDT3-D (Fig. 1C, G, K, O, S), it is
conceivable that some methane could diffuse into the water column  where it may be oxidized
by either aerobic or anaerobic oxidation processes (depending on the presence or absence of
oxygen, respectively) before emission into the atmosphere (Reeburgh, 2007). However,
benthic chamber incubations at the SBB stations did not indicate a release of methane into the
water column (Fig. S1), emphasizing the importance of cryptic methane cycling for preventing
the build-up of methane in the surface sediment and its emission into the water column.

*4.2. Rapid turnover of metabolic substrates*
Natural porewater MMA concentrations were mostly below detection (<3 µM);
however, in porewater close to the sediment-water interface of SDRO and NDT3-A, MMA
was detected but below the quantification limit (<10 µM) (Table S1). Although we are unable
to report definitive MMA concentrations, we can bracket the MMA concentrations in a range
between 3 and 10 µM. The bracketed MMA concentrations are about 1 to 2 orders of magnitude
higher than what has been reported from interstitial porewater at other locations. For example,
studies of sediment porewater off the coast of Peru found MMA concentrations to be ~0.15



μM (Wang and Lee, 1990). Similarly, in sediment porewater collected from Buzzards Bay,
Massachusetts and in the Eastern Tropical North Pacific Ocean, porewater MMA
concentrations were either present at trace amounts or below detection limit (<0.05 nmol g dry
wt[-1]) (Lee and Olson, 1984). Detectable but low methylamine concentrations in the porewater
found in our study could imply that methylamines are rapidly consumed by microbiological
processes and/or removed from the porewater through binding to minerals (Wang and Lee,
1990; Wang and Lee, 1993; Xiao et al., 2022). Our study provided support for both hypotheses
as we detected the biological potential for MMA consumption via radiotracer ([14]C-MMA)
experiments (Fig. 1) and detected the binding of 7-25% the injected [14]C-MMA to sediment (see

3.3.1).

Porewater methanol concentrations in the present study were also mainly below
detection, except for one sample, where it was not quantifiable (NDT3-A, 1–2 cm; Table S1).
In the marine environment, methanol is known to be a non-competitive substrate for
methanogenesis (King et al., 1983; Oremland and Taylor, 1978). However, a recent study
demonstrated that methanol is a carbon source for a wide variety of metabolisms, including
sulfate-reducing and denitrifying bacteria, as well as aerobic and anaerobic methylotrophs
(Fischer et al., 2021), which could all be present in the SBB sediments keeping methanol
concentrations low. Acetate was also detected in the metabolomic analysis but mostly below
quantification (except NDT3-A, 1–2 cm; Table S1). Acetate is formed through fermentation
reactions or through homoacetogenesis (Jørgensen, 2000; Ragsdale and Pierce, 2008). It is a
favourable food source for many bacteria and archaea such as sulfate reducers and
methanogens (Jørgensen, 2000; Conrad, 2020), which would explain its low concentration in
the SBB sediments. Low concentrations of the abovementioned metabolites are likely
signatures of rapid metabolic turnover, similar to what has been described for microbial
utilization of hydrogen in sediment (Conrad, 1999; Hoehler et al., 2001). In this situation,



metabolites would be kept at a steady-state concentration close to the thermodynamic
equilibrium of the respective consumers.

### 4.3. Competitive methylamine turnover by non-methanogenic pathways

Large disparities were found between AOM rates determined from the direct injection

of $^{14}C\text{-}CH_4$ (i.e., AOM-CH$_4$) and AOM determined from the production of $^{14}C$-TIC in the $^{14}C$-
MMA incubations (i.e., AOM-MMA). AOM-CH$_4$ was roughly 1-2 orders of magnitude lower
compared AOM-MMA (compare Fig. 1 B/C, F/G, J/K, N/O, R/S), indicating that AOM rates
determined via $^{14}C$-MMA incubations were overestimated. We hypothesize that this disparity
is the result of the direct conversion of $^{14}C$-MMA to $^{14}C$-TIC by processes other than AOM
coupled to MG-MMA. Any process converting $^{14}C$-MMA directly to $^{14}C$-TIC would inflate
the rate constant only slightly for MG-MMA, but dramatically for AOM-MMA (see Eq. 8, 9,
and 10). Fig. 1D, H, L, P, and T confirm that the rate constants for AOM-MMA are 1 to 2
orders of magnitude higher compared to AOM-CH$_4$ and MG-MMA. The difference in rate
constants strongly suggests that the $^{14}C$-TIC detected in the analysis of samples incubated with
$^{14}C$-MMA must result not only from AOM involved in the cryptic methane cycle but also from
direct methylamine oxidation by a different anaerobic methylotrophic metabolism that could
not be disambiguated using the adapted radiotracer method.

Methylamines are the simplest alkylated amine derived from the degradation of choline

and betaine found in plant and phytoplankton biomass (Oren, 1990; Taubert et al., 2017). The
molecules are ubiquitously found in saline and hypersaline conditions in the marine
environment (Zhuang et al., 2016; Zhuang et al., 2017; Mausz and Chen, 2019). The
importance of methylamine as a nitrogen and carbon source for microbes to build biomass has
been well documented (Taubert et al., 2017; Capone et al., 2008; Anthony, 1975; Mausz and
Chen, 2019). Methylamines can be metabolized by aerobic methylotrophic bacteria (Taubert
et al., 2017; Chistoserdova, 2015; Hanson and Hanson, 1996) and by methylotrophic



methanogens anaerobically (Chistoserdova, 2015; Thauer, 1998). Here we hypothesize that, in
addition to methylotrophic methanogenesis, sulfate reduction was involved in MMA
consumption in surface sediment of the SBB.

Recent literature does implicate anaerobic methylamine oxidation by sulfate reduction.

For example, Cadena et al. (2018) performed in vitro incubations with microbial mats collected
from a hypersaline environment with various competitive and non-competitive substrates
including tri-methylamine. Microbial mats incubated with trimethylamine stimulated
considerable methane production; but after 20 days, $H_2S$ began to accumulate and plateaued
after 40 days, suggesting that trimethylamine is not exclusively shuttled to methylotrophic
methanogenesis. The molecular data reported in Cadena et al. (2018), however, could not
identify a particular group of sulfate-reducing bacteria that proliferated by the addition of
trimethylamine. Instead, their molecular data suggested potentially other, non-sulfate reducing
bacteria, such as those in the family *Flavobacteriaceae* to be responsible for trimethylamine
turnover.

Zhuang et al., (2019) investigated heterotrophic metabolisms of C1 and C2 low

molecular weight compounds in anoxic sediment collected in the Gulf of Mexico. Sediment
was incubated with a variety of $^{14}C$ radiotracers alone and in combination with molybdate, a
known sulfate reducer inhibitor, to elucidate the metabolic turnover of low molecular weight
compounds, including $^{14}C$-labeled trimethylamine. Their results showed that although
methylamines did stimulate methane production, radiotracer incubations with molybdate and
methylamine demonstrated the inhibition of direct oxidation of $^{14}C$-methylamine to $^{14}C$-$CO_2$,
suggesting that methylamines were simultaneously oxidized to inorganic carbon by non-
methanogenic microorganisms. This finding further suggests a competition between
methanogens and sulfate-reducing bacteria for methylamine; however, the authors could not
rule out AOM as a potential contributor to the inorganic carbon pool.



Kivenson et al., (2021) discovered dual genetic code expansion in sulfate-reducing

bacteria from sediment within a deep-sea industrial waste dumpsite in the San Pedro Basin,
California, which potentially allows the metabolization of trimethylamine. The authors
expanded their study to revisit metagenomic and metatranscriptomic data collected from the
Baltic Sea and in the Columbia River Estuary and found expression of trimethylamine
methyltransferase in Deltaproteobacteria. This result suggested that a trimethylamine
metabolism does exist in sulfate-reducing bacteria which was enabled by the utilization of
genetic code expansion. Furthermore, the results also suggest that trimethylamine could be the
subject of competition between sulfate-reducing bacteria and methylotrophic methanogens.

Although the evidence of sulfate-reducing bacteria playing a larger role in methylamine

utilization is growing, there are other methylotrophic microorganisms in anaerobic settings that
could also be responsible for degrading methylamines. De Anda et al. (2021) discovered and
classified a new phylum called Brockarchaeota. The study reconstructed archaeal metagenome-
assembled genomes from sediment near hydrothermal vent systems in the Guaymas Basin,
Gulf of California, Mexico. Their findings showed that some Brockarchaeota are capable of
assimilating trimethylamines, by way of the tetrahydrofolate methyl branch of the Wood-
Ljunghal pathway and the reductive glycine pathway, bypassing methane production in anoxic
sediment.

Farag et al. (2021) found genomic evidence of a novel Asgard Phylum called

*Sifarchaeota* in deep marine sediment off the coast of Costa Rica.  The study used comparative
genomics to show a cluster, *Candidatus* Odinarchaeota within the *Sifarchaeota* Phylum, which
contains genes encoding for an incomplete methanogenesis pathway that is coupled to the
carbonyl branch of the Wood-Ljunghal pathway. The results suggest that this cluster could be
involved with utilizing methylamines. The *Sifarchaeota* metagenome-assembled genomes
results found genes for nitrite reductase and sulfate adenylyltransferase and phosphoadenosine
phosphosulfate reductase, indicating *Sifarchaeota* could perform nitrite and sulfate reduction.



But their study did not directly link nitrite and sulfate reduction to the utilization of
methylamines by *Sifarchaeota*.
Molecular analysis was not performed in the present study; therefore, we are unable to
directly link sulfate-reducing or any other heterotrophic bacteria to the direct anaerobic
oxidation of methylamine in the SBB. Future work should combine available geochemical and
molecular tools to piece together the complexity of metabolisms involved with methylamine
turnover and how it may affect the cryptic methane cycle. We note that there appears to be a
growing paradigm shift in the understanding of the utilization of non-competitive substrates in
anoxic sediment by sulfate-reducing bacteria and methylotrophic methanogens (including
other supposedly non-competitive methanogenic substrates like methanol (Sousa et al., 2018;
Fischer et al., 2021)). Apparently, methanogens are in fact able to convert these substrates into
methane in the presence of their competitors. Which factors provide them this capability should
be the subject of future research.

### 4.4. Implications for cryptic methane cycling in SBB

The SBB is known to have a network of hydrocarbon cold seeps, where methane and
other hydrocarbons are released from the lithosphere into the hydro- and atmosphere either
perennially or continuously (Hornafius et al., 1999; Leifer et al., 2010; Boles et al., 2004). The
migration of methane and other hydrocarbons vertically into the hydrosphere occur along
channels that are focused and permeable, such as fault lines and fractures (Moretti, 1998;
Smeraglia et al., 2022). Local tectonics and earthquakes could create new fault lines or fractures
that reshape or redisperse less permeable sediments, which may open or close migration
pathways for hydrocarbons, including methane (Smeraglia et al., 2022). In fact it has been
shown that hydrocarbons move much more efficiently through faults when the region in
question is seismically active on time scales <100000 yrs (Moretti, 1998). Given the current
and historical seismic activity (Probabilities, 1995) and faulting (Boles et al., 2004) within and



surrounding the SBB, it is conceivable that hydrocarbon seep patterns and seepage pathways
could also shift over time. A potential consequence of this shifting in the SBB is that methane
seepage could spontaneously flow through prior non-seep surface sediment. The fate of this
methane would then fall on the methanotrophic communities that are part of the cryptic
methane cycle. However, it is not well understood how quickly anaerobic methanotrophs could
handle this shift due to their extremely slow growth rates (Knittel and Boetius, 2009; Wilfert
et al., 2015; Nauhaus et al., 2007; Dale et al., 2008a). After gaining a better understanding of
cryptic methane cycling in the SBB presented in this study, a hypothesis worth testing in future
studies is whether cryptic methane cycling based on methylotrophic methanogenesis primes
surface sediments to respond faster to increases in methane transport through the sediment.



## 5. Conclusions

In the present study, we set about to find evidence of cryptic methane cycling in the sulfate-reduction zone of sediment along a depth transect in the oxygen-deficient SBB using a variety of biogeochemical analytics. We found that, within the top 10-20 cm, low methane concentrations were present within sulfate-rich sediment and in the presence of active sulfate reduction. The low methane concentrations were attributed to the balance between methylotrophic methanogenesis and subsequent consumption of the produced methane by AOM. Our results therefore provide strong evidence of cryptic methane cycling in the SBB. We conclude that this important, yet overlooked, process maintains low methane concentrations in surface sediments of this OMZ, and future work should consider cryptic methane cycling in other OMZ's to better constrain carbon cycling in these expanding marine environments.

Our radiotracer analyses further indicated microbial activity that oxidizes monomethylamine directly to $CO_2$ thereby bypassing methane production. Based off the sulfate reduction activity and methylamine consumption to $CO_2$ detected in this study and the metagenomic clues presented in the literature, we hypothesize that sulfate reduction may also be supported by methylamines. Our study highlights the metabolic complexity and versatility of anoxic marine sediment near the sediment-water interface within the SBB. Future work should consider how methylamines are consumed by different groups of bacteria and archaea, how methylamine utility by other anaerobic methylotrophs affects the cryptic methane cycle and evaluate if potential environmental changes affect the cryptic methane cycle activity.



**Data Availability Statement**

Porewater sulfate concentrations and sulfate reduction rates are accessible through the Biological & Chemical Oceanography Data Management Office (BCO-DMO) under the following DOI's:

http://dmoserv3.bco-dmo.org/jg/serv/BCO-DMO/BASIN/porewater_geochemistry.html0,

http://dmoserv3.bco-dmo.org/jg/serv/BCO-DMO/BASIN/sediment_parameters.html0,

http://dmoserv3.bco-dmo.org/jg/serv/BCO-DMO/BASIN/microbial_activity.html0.

Sediment methane concentrations and rates and rate constant data of AOM and methanogenesis can be found in the supplementary material Table S2.

**Author Contributions**

SK and TT designed the study; SK, JL, DY, DR, DH, QQ, FW, and FJ performed experiments and made measurements; SK, JL, DY, DR, DH, QQ, FW, FJ, DV, and TT analysed the data; SK and TT wrote the manuscript draft with input from all co-authors.

**Competing Interests**

Some authors are members of the editorial board of Biogeoscience. The peer-review process was guided by an independent editor, and the authors have also no other competing interests to declare.

**Acknowledgements**

We thank the captain and crew of R/V Atlantis, the crew of ROV Jason, the crew of AUV Sentry, and the science party of the research cruise AT42-19 for their technical and logistical support. This work was supported by the National Science Foundation NSF Award NO.: EAR-1852912, OCE-1829981 (to TT), and OCE-1830033 (to DV).



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
