# Peer review of "Evidence of cryptic methane cycling and non-methanogenic"

_EGUsphere, 2023_

## Author Response (AR2)

**Referee Comments**

**Author Response**

**Referee 1-Review**

**RC1 Comment:** This study "*Evidence of cryptic methane cycling and non-methanogenic methylamine consumption in the sulfate-reducing zone of sediment in the Santa Barbara Basin, California*" presents evidence of cryptic methane cycling activity within the upper regions of the sulfate-reducing zone, along a depth transect within the Santa Barbara Basin by the combination of geochemical analysis, metabolic analysis and radio-tracer labelling. The topic of cryptic methane cycling is of high interest to this field, the data is of good quality and the manuscript is well written. I find the topic of this study certainly interesting and suitable for this journal, and there are only a few minor points which preclude its publication at this moment.

**Authors' Response**: We would like to thank the referee for the positive critique on the present manuscript. Below we address the comments and edits from the referee.

**Minor**

**RC1 Comment:** Line 119 make

**Authors' Response:** Thank you for catching the typo. We plan to correct this in the new version of the manuscript.

**RC1 Comment:** Line 148 stations'

**Authors' Response**:

Thank you for catching the typo. We plan to correct this in the new version of the manuscript.

**RC1 Comment:** Line 189 7 to seven

**Authors' Response**: Thank you for catching the typo. We plan to correct this in the new version of the manuscript.

**RC1 Comment:** Line 223 would not call an NMR based method for porewater concentrations of acetate, methanol.. a metabolomic analysis. Rather It usually means analyzing proteins.

**Authors' Response**: Thanks to the referee for the comment. We are of course focused on the mixtures of small organics (from pore water directly (centrifuged) not dried down) that can be identified by performing spectral library matching (screening over 1000 potential metabolites) in spectra that were collected - in the same way as these spectra. Acetate, methanol etc. at their natural concentrations. Efforts to "dry down" samples and look for a longer list of non-volatile organics (amino acids, sugars etc.) is possible but not done here as the focus for this study is on the fermentation products, etc., which could be altered by drying down process (lyophilization or

speed -vac etc.). The spectral optimization is for the small molecular weight molecules (generally less than 300 amu) - highly mobile (and highly resolved) peaks - so not the proteins in this case (broader lines and at natural concentrations not high enough to produce significant contributions to any of the spectra). Further, quantification of the small metabolites reported (acetate, methanol, etc.) is accomplish by incorporation of a known quantity of an internal standard (NIST certified) in to the sample that matches the collection conditions of the largely commercially developed library (Cheneaux).

However, we do see the referee's point on clarity of the title. Since the present manuscript is focused on methylotrophic methanogenesis and thus interested in the presence and availability of methanogenic substrates we have changed the chapter title to be more in line with the focus of the paper. We have changed the title to read "***Determination of methanogenic substrates in porewater***"

**RC1 Comment:** Line 289 Eq. 7

**Authors' Response**: Thank you for catching the typo. We plan to correct this in the new version of the manuscript.

**RC1 Comment:** Line 319 What's the point of defining a seemingly strange rate constant here? Any reference?

**Authors' Response**: Thank you for the comment. We define the rate constant $k$ in section 2.5.4 because they are plotted in Figure 1 to show how they compare. More importantly the rate constants provide evidence that $^{14}$C-mono-methylamine is consumed by a non-methanogenic process and inflate the ex-situ rates for AOM-MMA. We discuss this in detail in discussion sections 4.2 and 4.3. We plan to add information at the beginning of the method section to clarify the purpose of the constant calculations.

**RC1 Comment:** Line 430-432 what does this mean exactly

**Authors' Response**: Thank you to the referee for the comment. We realize this sentence was not written clearly. We plan to rewrite the sentence in the new version of the manuscript as follows: "*This pattern suggests that the partial pressure of methane was likely at thermodynamic equilibrium between methanogenesis and AOM (compare, e.g., with Conrad 1999).*"

**RC1 Comment:** Line 508-510 It is not a suitable or should not be called a hypothesis in discussion part. Moreover, the hypothesis is not tested with strong proof either from literature or this study.

**Authors' Response**: Thank you for the comment we plan to change the phrasing accordingly in the new version:

**Referee Comments**

**Author Response**

Referee 2 - Review

**RC2 Comment:** The manuscript by Krause et al. presents compelling evidence for cryptic methane cycling and methylamine consumption in sediments from Santa Barbara Basin (SBB). This is a very well-designed study and the results are presented in a straightforward way. I have absolutely nothing to criticize on the methods, except perhaps that the radiotracer rate measurements were not performed in triplicates. Such measurements usually have a notoriously high variability, even on a very small spatial scale. In most

**Authors' Response:** We would like to thank the referee for reading the manuscript and providing positive critiques. Below we address all the referee's comments and/or edits of the present manuscript.

**RC2 Comment:** Therefore, I am wondering why no replicates were taken. There might not have been more cores available, but at least this should be mentioned somewhere.

**Authors' Response**: Thank you for the comment. The referee is of course right that spatial heterogeneity contributes to variability in the ex-situ rates that we are interested in. The reason we did not do replicate radiotracer experiments is in fact because not enough sediment cores were available to perform replicate incubations. We plan to add a short sentence at the end of section 2.5 of the new version to clarify this.

Minor comments:

**RC2 Comment:** line 97: what do you mean by "myriad environments"? This phrase was already marked yellow in the pdf, perhaps someone else was questioning it already?

**Authors' Response**: Thank you, we plan to remove the word "myriad" from the sentence and reworded the sentence to be clearer in the new version.

**RC2 Comment:** line 107: remove "strong", any oxygen fluctuation can cause these conditions

**Authors' Response**: Thank you, we agree. We plan to remove the word "strong" in the new version of the manuscript.

**RC2 Comment:** line 122: remove "the" before coastal marine environments

**Authors' Response**: Thank you. We plan to correct this in the new version of the manuscript.

**RC2 Comment:** line 126: …within OMZ's is virtually unknown

**Authors' Response**: Thank you. We plan to correct this in the new version of the manuscript.

**RC2 Comment:** line 180: Isn't this company called "Scott Specialty Gases" since a few years? I know the old term Scotty, but I think that has changed since they were taken over by another company

**Authors' Response**: Thank you. We plan to correct this in the new version of the manuscript.

**RC2 Comment:** line 196-206: That part should be restructured. At first I was at loss why there are suddenly 26 ml serum bottles, but then in line 203 you mention that they are used for the water from the syringes. It would make it easier for the readers if you could mention the use of the serum bottles in the first sentence and then describe how they are prepared and how the water is transferred.

**Authors' Response**: Thank you for pointing this out. We plan to restructure the paragraph to be clearer in the new version of the manuscript.

**RC2 Comment:** line 252: As mentioned before, why did you take only three mini cores for the three processes. i.e no replicates?

**Authors' Response**: Thank you for the comment. As said previously, the referee is of course right that spatial heterogeneity contributes to variability in the ex-situ rates that we are interested in. The reason we did not do replicate radiotracer experiments is in fact because not enough sediment cores were available to perform replicate incubations. We plan to add a short sentence at the end of section 2.5 of the new version to clarify this.

**RC2 Comment:** line 336: add a komma after "trends"

**Authors' Response**: Thank you. We plan to correct this in the new version of the manuscript.

**RC2 Comment:** line 363f: Why are the SRR for the top 5 cm missing? You only state that they are, which is visible on the figure, but an explanation would be appreciated.

**Authors' Response**: SRR data are missing in the top 5 cm at SDRO is due to post-cruise analytical issues. We plan to rewrite the sentence in the new version to clarify the reason for the missing SRR data.

**RC2 Comment:** lines 433 and 436: In one sentence you write that the methane concentrations are "relatively constant", then a bit later you state that there are "non-linear" methane trends. That is somewhat confusing.

**Authors' Response**: Thank you, we plan to reword the sentence for better consistency in the wording.

**RC2 Comment:** line 440: I suggest rephrasing the start of the sentence to "As there is considerable methanogenic activity even at the……"

**Authors' Response**: Thank you for the suggestion, we will incorporate this into the new version of the manuscript.

**RC2 Comment:** line 458: Remove "porewater", it is already mentioned before

**Authors' Response**: Thank you. We plan to correct this in the new version of the manuscript.

**RC2 Comment:** line 459: Could you convert the concentration to µM, so it would be easier to compare to the value mentioned before?

**Authors' Response**: Thank you, we will convert the concentration to µM for better comparison in the new version.

**RC2 Comment:** line 489: compared to AOM-MMA

**Authors' Response**: Thank you. We plan to correct this in the new version of the manuscript.

**RC2 Comment:** line 494-494: Why is that? This sentence is a statement, but I don't see an explanation for it.

**Authors' Response**: The interpretation of the statement is actually in the following sentence.

**RC2 Comment:** line 500f. This sentence is a bit hard to follow. I would suggest splitting it: "Methylamines are the simplest alkylated amines. They are derived from the degradation of choline and betaine and found in plant and phytoplankton biomass (citation)."

**Authors' Response**: Thank you for the suggestion, we plan to adopt it in the new version.

**RC2 Comment:** line 549 and 555: The pathway is called "Wood-Ljungdahl"

**Authors' Response**: Thank you. We plan to correct this in the new version of the manuscript.

**RC2 Comment:** line 559: Replace "But" by "However, "

**Authors' Response**: Thank you. We plan to correct this in the new version of the manuscript.

**Associate Editor Decision**

Dear Dr. Krause

Thank you for submitting your revised version to Biogeosciences. I have read it with pleasure and I am happy to inform you that your paper is now accepted for publication, pending a few minor technical corrections.

With best regards, Jack Middelburg, Associate Editor

**Authors Response:** We would like to thank the Associate Editors efforts in facilitating the peer-review process for this manuscript and for their careful review of the manuscript. Below we address the Associate editors' technical edits to the manuscript.

**Associate Editor Comments:**

Line 356: depict rather than depicts

**Authors response:** Thank you for identifying the typo. This is now fixed in the newest version.

Line 385: show rather than shows

**Authors response:** Thank you for identifying the typo. This is now fixed in the newest version.

Line 459: interstitial porewater replace by either interstitial water or porewater.

**Authors response:** Thank you we have removed the word "interstitial" from the newest version of the manuscript.